# Capsaicinoids and Their Effects on Cancer: The “Double-Edged Sword” Postulate from the Molecular Scale

**DOI:** 10.3390/cells12212573

**Published:** 2023-11-04

**Authors:** Francisco Luján-Méndez, Octavio Roldán-Padrón, J. Eduardo Castro-Ruíz, Josué López-Martínez, Teresa García-Gasca

**Affiliations:** 1Laboratorio de Biología Celular y Molecular, Facultad de Ciencias Naturales, Universidad Autónoma de Querétaro, Av. De las Ciencias s/n, Juriquilla, Querétaro 76230, Querétaro, Mexico; francisco.lujan@uaq.mx (F.L.-M.); octavio_rolpad@hotmail.com (O.R.-P.); francisco.josue.lopez@uaq.mx (J.L.-M.); 2Escuela de Odontología, Facultad de Medicina, Universidad Autónoma de Querétaro, Querétaro 76176, Querétaro, Mexico; jesus.castro@uaq.mx

**Keywords:** apoptosis, autophagy, vanilloid-like transient potential receptors (TRPV), tumor-associated NADH oxidase (tNOX), reactive oxygen species (ROS), silent mating-type information regulation 2 homolog 1 (SIRT1), p53, epithelial–mesenchymal transition (EMT), immunogenic cell death (ICD), tumor microenvironment (TME)

## Abstract

Capsaicinoids are a unique chemical species resulting from a particular biosynthesis pathway of hot chilies (*Capsicum* spp.) that gives rise to 22 analogous compounds, all of which are TRPV1 agonists and, therefore, responsible for the pungency of *Capsicum* fruits. In addition to their human consumption, numerous ethnopharmacological uses of chili have emerged throughout history. Today, more than 25 years of basic research accredit a multifaceted bioactivity mainly to capsaicin, highlighting its antitumor properties mediated by cytotoxicity and immunological adjuvancy against at least 74 varieties of cancer, while non-cancer cells tend to have greater tolerance. However, despite the progress regarding the understanding of its mechanisms of action, the benefit and safety of capsaicinoids’ pharmacological use remain subjects of discussion, since CAP also promotes epithelial–mesenchymal transition, in an ambivalence that has been referred to as “the double-edge sword”. Here, we update the comparative discussion of relevant reports about capsaicinoids’ bioactivity in a plethora of experimental models of cancer in terms of selectivity, efficacy, and safety. Through an integration of the underlying mechanisms, as well as inherent aspects of cancer biology, we propose mechanistic models regarding the dichotomy of their effects. Finally, we discuss a selection of in vivo evidence concerning capsaicinoids’ immunomodulatory properties against cancer.

## 1. Introduction

In 1876, the English pharmacist, physician, and public health specialist John Clough Thresh isolated capsaicin, the main pungent compound of hot peppers (*Capsicum* spp.), for the first time. After 147 years, research to elucidate the spectrum of its bioactivity continues at a historic pace [1] in order to assess the exploitation of the nutraceutical and pharmacological potential of these compounds.

Capsaicinoids are a group of secondary metabolites of the genus *Capsicum*, and more than 22 analogous compounds from its fruits have been identified [2], highlighting capsaicin (8-Methyl-N-vanillyl-trans-6-nonenamide; CAP) and dihydrocapsaicin (8-Methyl-N-vanillylnonanamide; DHC), which typically represent between 80 and 90% of the total capsaicinoids [3,4]. These compounds are ligand agonists of vanilloid-like transient potential receptor 1 (TRPV1), a transmembrane protein expressed in multiple tissues, whose function in sensory neurons is associated with thermal and neuropathic hyperalgesia as well as the perception of the sensation of pain and burning (pungency) inherent to the consumption of hot peppers [5]. The human race has a history of exposure to these compounds that goes back 6000 years to the peoples of pre-Columbian America, from whose archaeological remains have made it possible to identify chili starch microfossils in objects such as grinding stones and ceramic fragments [6]. To this day, the characteristic pungency of *Capsicum* fruits remains an essential trait in a variety of cuisines [7]. Parallel to their use as food, multiple ethnopharmacological uses of *Capsicum* species have emerged throughout history; for example, the botanical pharmacopoeia of the Mayan culture proposes the use of *Capsicum* species in remedial preparations for respiratory and intestinal problems, burns, ear pain, and infected wounds [8]. Less than 200 years after their discovery by Europeans, the five domesticated *Capsicum* species were successfully introduced to the rest of the world and, therefore, increased the range of their medicinal uses [9].

*Capsicum* fruits are a source of a variety of metabolites of importance for human health [10,11,12]. Their content of carotenoids [13], ascorbic acid [14], tocopherols, phenols [15,16], capsaicinoids [17,18], and, to a lesser extent, other compounds [19,20] has been proposed to validate some of the ethnopharmacological uses of chili [21,22]. Regarding its consumption, a 9-year longitudinal observation of 485,000 adults aged between 30 and 79 years, resident in 10 different regions of China, revealed an inverse association between the self-reported frequency of consumption of foods derived from *Capsicum* spp. and the total mortality in terms of the absolute rates observed in three consumption categories: 1 or 2 times per week (with 4.4 deaths per 1000 individuals per year), 3 to 5 times (with 4.3 deaths), and 6 to 7 times (with 5.8 deaths). These were compared to individuals who consumed the foods less than once per week (6.1 deaths), as well as by calculating the adjusted hazard ratios for death, which were 0.90, 0.86, and 0.86, respectively, in each category (95% confidence interval). After performing a multivariate adjustment, inverse associations were also detected between the consumption of foods derived from *Capsicum* spp. and the risk of specific mortality due to a variety of causes including cancer, ischemic heart disease, and respiratory diseases [23]. Shortly before, a similar association between overall mortality and hot pepper consumption was reported in a representative sample of noninstitutionalized adults in the United States over an observation period of six years, with a 12% reduction in the absolute mortality rate in the participating consumers and a multivariate-adjusted hazard ratio of 0.87 (95% confidence interval) [24]. In this context, capsaicinoids have been implicitly pointed out, since both the degree of pungency and the frequency of consumption of spicy foods were inversely associated (*p* < 0.05) with serum LDL cholesterol concentrations and the LDL cholesterol: HDL cholesterol ratio in a Chinese population (N = 6774) aged 18 to 65 years, without a diagnosis of pre-existing cardiovascular diseases, through multilevel mixed-effects models adjusted for confounders and cluster effects. In addition, those individuals who consumed spicy foods more than 5 times per week presented positive correlations with HDL cholesterol concentrations (*p* < 0.01) [25]. Therefore, even though the non-participant research does not support inferences of causality, the notion emerges that the contribution of capsaicinoids to human health could be significant. In this regard, a growing body of evidence of the consumption of certain foods and the subsequent reduction in some parameters of oxidative damage has prompted considerable interest in the identification and functional characterization of phytochemicals synthesized in *Capsicum* fruits and other vegetables, in clear orientation to their evaluation as drug candidates [26].

Capsaicinoids, and CAP in particular, have been extensively studied, and various biological properties of pharmacological relevance have been postulated, such as lipid-lowering, antilithogenic, antioxidant, analgesic, antidiabetic, anti-inflammatory, antiulcer, anti-obesogenic and anticancer activities. Within the multifaceted bioactivity of capsaicin (which has been the subject of numerous reviews [27,28,29,30,31,32,33,34]), its potential as an anticancer agent is possibly one of the fields that has aroused the most interest over the years. The antiproliferative and apoptotic effects of this capsaicinoid on cancer cell lineages denote remarkable selectivity, as CAP tolerance in normal cell systems is significantly higher, as observed in a wide variety of in vitro reference models including both primary cell cultures of rat and human hepatocytes [35,36], human astrocytes [37], and cells derived from the normal small airway epithelium (SAEC), bronchial/tracheal epithelium (NHBE) [38], pancreatic duct epithelium (HPDE6-E6E7) [39], human fetal lung fibroblasts (MRC-5) [40], mouse embryonic fibroblasts (MEFS) [41,42] and human embryonic kidney cells (HEK-293) [43].

However, despite the growing body of evidence on the various therapeutic potentials, the benefit and safety of the medical use of capsaicinoids remains subject to discussion due to contrasting data, particularly on cancer [44]. A series of works indicates that, in contrast to its widely postulated antitumor properties, under certain experimental conditions, CAP promotes the development of aggressive cancer phenotypes in an ambivalence that has come to be referred to in the literature as a “double-edged sword” and “the two faces of capsaicin” [45,46]. Here, we start from the biosynthesis routes of these compounds to discuss the most recent and also the most classical reports regarding their bioactivity in a wide variety of experimental models of cancer in terms of their selectivity, efficacy, and safety. Based on underlying molecular events, described by works on both sides of the spectrum as well as inherent aspects of the biology of certain cancers, we propose mechanistic models. Then, we highlight a selection of in vivo evidence regarding capsaicinoids’ antitumor activity and their immunomodulatory properties against cancer.

## 2. Capsaicinoids: In Situ Synthesis and Bioactivity

Taxonomically, *Capsicum* spp. belong to the Solanaceae family, the Asteridae clade of the eudicots, and constitute diploid, facultative, and self-pollinating organisms [47]. Of the more than 30 categorized *Capsicum* species, only 5 have become domesticated crops—*Capsicum annuum*, *Capsicum frutescens*, *Capsicum baccatum*, *Capsicum chinense*, and *Capsicum pubescens* [12,48]—and, as such, are the most relevant sources of capsaicinoids in nature.

Capsaicinoids are water-insoluble, odorless, and colorless compounds, synthesized in the epidermis of the placenta of *Capsicum* fruits by cellular structures called glands or blisters, and then stored in vesicles (0.15 to 1.0 µm in diameter) typically visible on the surface of the tissue between 15 and 31 days post anthesis according to the species [49,50].

Biosynthetically, capsaicinoids are the product of the convergence of two pathways (Figure 1): the phenylpropanoid pathway that provides vanillylamine, and the metabolism of branched-chain amino acids that participate with different branched fatty acids (acyl-fatty derivatives) from 9 to 11 carbon atoms in length and a variable number of double bonds at different points in the chain. A capsaicinoid emerges whenever a fatty acid is linked by the enzymatic condensation of an amide group to the benzene ring of vanillylamine, which constitutes its phenolic portion [51,52].

The acyltransferase responsible for the condensation of vanillylamine with some of the available fatty acids is capsaicin synthase [11], and, as such, it represents an attractive biotechnological target for the modulation of capsaicinoid synthesis in situ [53]. However, the other enzymes involved are not properly characterized and the signaling pathway is not fully understood, which is why it remains an object of study today [49,54,55].

**Figure 1 cells-12-02573-f001:**
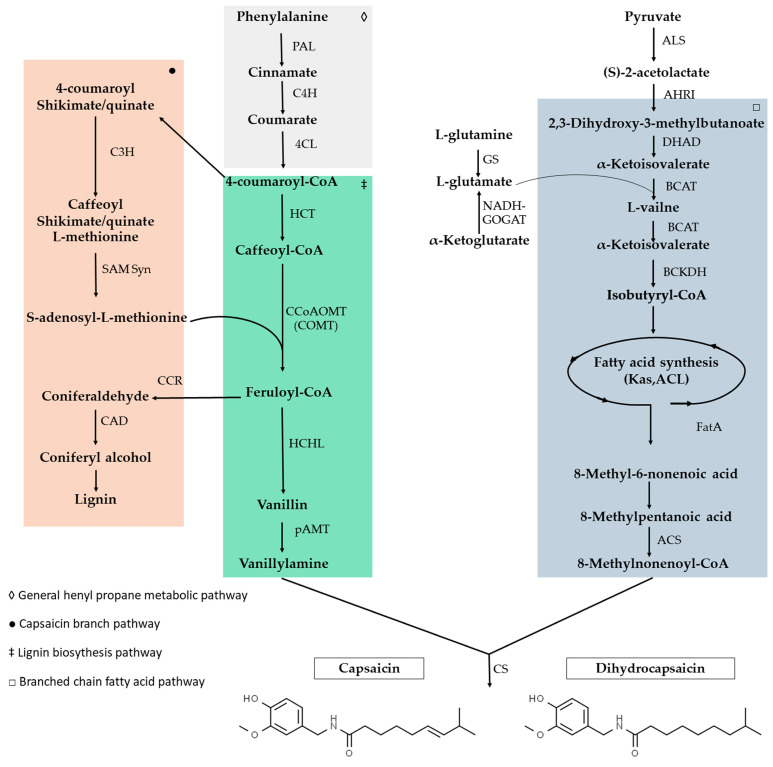
Main biosynthetic pathway proposed for capsaicin, showing the convergence of the phenylpropanoid pathway and the metabolism of the branched-chain amino acid route. PAL, phenylalanine ammonia lyase; C4H, cinnamate 4-hydroxylase; 4CL, 4-coumaroyl-CoA ligase; C3H, coumaroyl shikimate/quinate 3-hydroxylase; HCT, hydroxycinnamoyl transferase; CCoAOMT, caffeoyl-CoA 3-O-methyltransferase; SAM Syn, S-adenosyl L-methionine synthase; COMT, caffeic acid O-methyltransferase; CCR, cinnamoyl-CoA reductase; CAD, cinnamylalcohol dehydrogenase; HCHCL, hydroxycinnamoyl-CoA hydratase/lyase; pAMT, putative amino transferase; GS, glutamine synthetase; GOGAT, glutamate synthase; ALS, acetolactate synthase; AHRI, acetohydroxy-acid reductoisomerase; DHAD dihydroxy acid dehydratase; BCAT, branched-chain amino acid aminotransferase; BCKDH, branched-chain α-keto-acid dehydrogenase complex; FatA, acyl-ACP thioesterase; ACS, acyl-CoA synthetase; CS, capsaicinoid synthase. Modified from [56,57,58].

It has been pointed out that capsaicinoids perform various functions in the plant, such as increasing its resistance to biotic stress due to its antifungal and antibacterial activity [59] and by discouraging their predation by mammals through their pungency [60]. Capsaicinoids also are related to the dispersal of seeds over long distances by birds [61], whose TRPV1 is not responsive to capsaicinoids due to a point difference in its vanilloid-binding pocket [47,62] at the level of Ala578 [63]. The pungency of chili capsaicinoids is the result of their agonism over TRPV1 in the membranes of a subpopulation of sensory neurons where, under a homotetrameric arrangement, they are responsible for transmitting information to the central nervous system in response to noxious stimuli of a diverse nature [64]. Such versatility is partly explained by the polymodal character with which TRPV1 performs its function as a non-selective cation channel in response to a wide spectrum of physical and chemical stimuli, such as painful heat, extracellular acidosis, and certain ligands; among these, CAP stands out, given that TRPV1 exhibits a submicromolar affinity not shared by the homologous channels TRPV2–TRPV6 [65]. Although the pharmacophoric profile of CAP has been extensively addressed in numerous studies [66], the detailed description of its supramolecular interaction with TRPV1 is relatively recent [65].

The activation or phosphorylation of TRPV1, respectively mediated by agonists or by one of the kinases involved, leads to cell membrane depolarization through the influx of Na^2+^ and Ca^2+^, the latter being particularly important in signaling with outcomes of inflammation and pain [67]. However, the prolonged or repeated exposure to CAP desensitizes TRPV1 and reversibly defunctionalizes the afferent nociceptive fibers that express it, leading to a long-lasting refractory period [54,68]. Mechanistically, CAP-mediated analgesia is associated with the depletion of the neuropeptide substance-P in the terminals of certain nociceptors [69] and, at a subcellular level, with the activation of proteins such as calmodulin (CaM) and calcineurin (PPP3C), which, after the increase in the intracellular concentration of Ca^2+^, are, respectively enabled to desensitize TRPV1 by preventing its phosphorylation through competitive binding to the ATP pocket and by direct dephosphorylation in Thr370 in an inhibition-by-feedback fashion [70,71]. Although all capsaicinoids are effective TRPV1 agonists, they exhibit different degrees of pungency, and in this sense, CAP and DHC are the most potent and also the most abundant [72].

## 3. Molecular Mechanisms behind the Selective Cytotoxicity of Capsaicin in Cancer Cells

Over more than 25 years, CAP has been reported to provoke apoptotic and inhibitory in vitro effects in a variety of human cancer cells and in explanted assays into rodents, amounting to at least 74 cell lines derived from malignancies of various histological origins including 57 carcinomas (5 murine), 3 sarcomas (2 murine), 11 leukemias, and 3 lymphomas [44]. The described mechanisms are related to the modulation of a series of molecules involved in the transduction of various signals, whose outcomes include oxidative stress, cell cycle arrest, mitochondrial dysfunction, endoplasmic reticulum stress [73], and the inhibition of angiogenesis and metastasis of established tumors [74]. Contrary to supposition, with some exceptions, these effects occur independently of the abundance or availability of the TRPV1 receptor in tumor cells [75,76,77,78].

Although such selectivity of the cytotoxic effects of CAP has led to it being widely postulated as an anticancer agent, even today, its mechanistic understanding is partial. In the line of evidence, it has been observed that the preferential expression of certain molecules in cancer cells determines the level of their susceptibility to CAP, and in this regard, the role of tumor-associated nicotinamide adenine dinucleotide oxidase (tNOX) is probably the one with the best experimental verification. As a member of the family of growth-related plasma membrane hydroquinone oxidases, tNOX possesses biochemical aptitudes for thiol/disulfide exchange and for nicotinamide adenine dinucleotide phosphate (NADP^+^) oxidation [79]. Since tNOX is the product of an aberrant differentiation program and not of an oncogenic mutation [80,81], its identity as an oncofetal antigen is well established. Its expression has been documented, on the one hand, during the embryonic development of chickens and in human embryonic kidney cells (HEK-293), and, on the other, in various human cancers [82,83].

More than 20 different isoforms of tNOX are known, which, by alternative mRNA splicing, designate a variety of products with molecular weights from 34 to 94 kDa expressed with some specificity in a wide variety of human malignancies [82]. The ectoenzyme tNOX is permanently activated and, unlike the constitutive NADH oxidase Ecto-NOX disulfide-thiol exchanger 1 (ENOX1), does not require hormones and/or growth factors for this purpose [84]. This characteristic leads to a remodeling of NADH metabolism in that it directly affects the basal NAD^+^/NADH ratio (product/substrate of tNOX), sustainably elevating it in cancer cells. Physiologically, the elevation of the intracellular NAD^+^/NADH ratio constitutes a resource for the development of aggressive phenotypes in cells of multiple malignancies, whose expression of tNOX correlates positively with exacerbations of proliferation, survival, and tumor progression, according to a series of works through loss- and gain-of-function strategies [85,86,87,88].

A growing body of evidence indicates that CAP exposure effectively abrogates the free and membrane-associated isoforms of tNOX [39,89,90,91,92,93,94,95], either in the serum of patients with active cancer through supramolecular interactions [89] or in cancer cell cultures, through the depletion of its transcriptional regulator [83] POU domain, class 3 transcription factor 2 (POU3F2). More recently, a cellular thermal shift assay (CETSA) methodology has demonstrated that CAP also binds directly to tNOX and leads to its degradation by the ubiquitin–proteasome and autophagy–lysosome pathways in bladder cancer in vitro (T24 cells) [88,96]. In all cases, CAP-mediated tNOX inhibition is causally related to a decline in the intracellular NAD^+^/NADH ratio and subsequent increases in the generation of extramitochondrial oxidative stress, which, in addition to affecting the proliferative potential, negatively affects the activity of the coenzyme NAD^+^-dependent deacetylase sirtuin-1 (SIRT1) and configures a transductional arrangement that occurs selectively in cancer cells. In the representative case of lung cancer cells (line A549), Lee et al. (2015) demonstrated that after 24 h of exposure to CAP (200 µM), a downregulation in the expression of tNOX ensued, followed by a drop in the NAD^+^/NADH intracellular ratio and consequent affectations in the expression and activity of SIRT1, whose derogation was succeeded by increases in both the acetylation (Lys382) and the phosphorylation (Ser46) of the tumor suppressor p53, leading to a significant increase in apoptotic activity in tumor cells. In contrast, after the same treatment, MRC-5 human lung fibroblasts (tNOX^−^) experienced an opposite effect on the intracellular NAD^+^/NADH ratio, the elevation of which resulted in a substantial enhancement of SIRT1 activity and the subsequent decline in p53 acetylation levels, followed by the induction of cellular autophagy without affectation on the viability of non-tumor cells [40].

Consistently, TSGH-8301 and T24 bladder cancer cells experienced effects that confirm the involvement of the tNOX/SIRT1/p53 axis in the proapoptotic mechanisms of CAP. The authors reported that CAP concentrations of 100 and 200 µM caused the depletion of cyclin D1, followed by a pronounced inhibition of Cyclin-dependent kinase 4 (CDK4) and the consequent cessation of its inhibitory influence by hyperphosphorylation over the tumor suppressor retinoblastoma protein (pRb), whose stabilization was related to a significant increase in subpopulations under cell cycle arrest at the G1 phase [94]. In this work, the CAP-mediated inhibition of tNOX also led to a significant loss in cell migration, through abrogation of the phosphorylation of Focal Adhesion Kinase (FAK), an intracellular protein tyrosine kinase that acts as the main regulator in the assembly and disassembly (turnover) of macromolecular complexes rich in integrins, called Focal Adhesions (FA). Their succession at different points of the membrane determines both adhesion and integrin-dependent cell migration, and also of paxillin (pax), also essential for the assembly of FA in the leading front of the migratory cell [97]. Recently, new data from the same work group enriched the transductional observations made in T24 cells exposed to CAP (100 and 200 µM) by demonstrating that NAD^+^-dependent SIRT1 abrogation also leads to increases in c-Myc acetylation, compromising its interaction with the Max protein, in what constitutes the previously demonstrated mechanistic explanation for cyclin D1 depletion [88].

In agreement with these findings, Pramanik et al. (2014) demonstrated that the inhibitory influence exerted by CAP on SIRT1 is not limited to a few human cancers in vitro by adding two cell types of pancreatic adenocarcinoma (BxPC-3 and AsPC-1 cell lines) and a squamous cell carcinoma (L3.6PL cell line) to the list. After the treatment with CAP (100–200 µM) over 24 h, all cell types experienced drops in the expression of SIRT1, SIRT2, and SIRT3, concurrent with an increase in acetyltransferase CREB-binding proteins (CBP) associated with increases in the acetylation and subsequent phosphorylation (Ser256) of forkhead box transcription factor-class O (FOXO-1), dependent on the phosphorylation (Tyr183/Tyr185) of the c-Jun amino terminal kinase (JNK). The effect improved the transcriptional activation of FOXO-1 and favored the expression of the protein Bcl-2 Interacting Mediator of cell death (BIM), whose translation and phosphorylation (Ser 69) were followed by an increase in caspase-3 activation, PARP-1 cleavage, and the foreseeable consequences in terms of an increase in cell apoptosis of the three tumoral types [98]. However, the possible implication of the derogation of tNOX in the process was not evaluated in this work, and instead, JNK phosphorylation was associated with increases in the oxidative stress of the CAP-treated cells, which, in turn, was also collaterally reported by Lee et al. (2015) [40]. Finally, in this line of evidence, two recent works demonstrated that the mechanistic tNOX/SIRT1/p53 axis also underlies the inhibitory effects of CAP in two human cell types of tongue squamous cell carcinoma (SAS and HSC-3 cell lines) [96], one of human malignant melanoma (A-357 cell line), and one of mouse melanoma (B16–F10) [88], although in these four cancer cell types, cellular autophagy contributed at different levels to the final cytotoxicity. In summary, the data collected from 11 different lineages of cancer indicate that CAP (100–200 µM) acts to alternatively induce signaling pathways through SIRT1 with two distinct outcomes: survival autophagy in non-transformed cells, and the reversal of invasive phenotype, apoptotic, and autophagic death in cancer cells. This is constituted as the mechanistic axis best supported by the evidence regarding the cytotoxic selectivity of CAP, but there is also a divergence of results in other cancer models.

It was observed that the moderately differentiated human gastric adenocarcinoma cells of the TMC-1 line exposed to CAP maintained stables levels of tNOX expression and oxidative stress. Remarkably, TMC-1 cells showed tolerance to CAP concentrations typically effective in achieving tNOX abrogation (≥200 µM), whose levels did not present statistically significant changes after 72 h of exposure [83,95], which could indicate the existence of decisive upstream molecular events. In this regard, it has been observed that the inhibitory activity of CAP on cells of human gastric adenocarcinoma (AGS cell line) and two types of small cell lung cancer (DMS53 and DMS114) is dependent on the availability of the TRPV6 ion channel, whose depletion by siRNA methodology led to the CAP-mediated cessation of apoptosis [38,99], whereas the pharmacological inhibition of TRPV1 does not appear to affect apoptotic activity in these and many other human cancer cell lines, including cells from colorectal cancer (HT-29), prostate cancer (PC-3), two cell types of pancreatic neuroendocrine tumor (BON-1 and QGP-1), and rat glioma (C6) [75,76,77,100].

The TRPV6 epithelial channel is one of the 28 transient potential receptors (TRP) expressed by mammals. It belongs to the same subfamily as the TRPV1 channel, but, unlike the latter, it exhibits a high selectivity for Ca^2+^ (with P_Ca_/_Na_ > 100) [101]. In its constitutively active character, TRPV6 is regulated through a rather restricted pattern of expression and cellular distribution, mainly in placenta, prostate, pancreas, and small intestine [102,103], where it contributes to the transepithelial transport of Ca^2+^ [104,105,106]. Interestingly, multiple analyses of cancer surgical specimens and cancers from various cell lines have shown substantially higher TRPV6 expression in prostate, breast, thyroid, colon, and ovarian cancers relative to their noncancerous counterparts [107,108,109,110], which could suggest the participation of TRPV6 in the pathological process [111,112], favoring the hypothesis of its involvement in the selectivity of the inhibitory properties of CAP. Therefore, even though no direct TRPV6 agonists have been identified to date [106], given the extraordinary sensitivity with which TRPV1 reacts to its ligands, it could be reasonable to hypothesize that TRPV6 might retain some affinity for certain TRPV1 agonists. However, to the best of our knowledge, the probable mechanisms of CAP-TRPV6 supramolecular interaction remain unexplored, and as such, constitute an interesting target for in silico analysis and the CETSA approach.

## 4. Mechanisms Underlying Capsaicin-Induced Apoptosis in Cancer: Brief State of the Art

Concerning the cytotoxic effects of CAP, the currently available body of evidence regarding the molecular events that underlie its inhibitory and apoptotic effects in cancer cells could reach sufficiency for preclinical purposes. Although most of these data come from traditional methodologies (studying one molecular target, in one cell type at a time), their comparative integration allows the building of a general mechanistic perspective on CAP-exerted proapoptotic bioactivity in cancer cell systems. CAP exposure affects the abundance and stoichiometry of numerous proteins associated with proapoptotic complexes in a manner closely linked to p53 stabilization achieved after certain post-translational modifications. In most cancers tested in vitro, after 6–24 h of exposure to CAP, JNK phosphorylates p53 (Thr81), resulting in an impediment to its ubiquitylation by the murine double minute protein 2 (Mdm2), and, therefore, its subsequent degradation by the proteasome system. Thus, the half-life of p53 is substantially improved, followed by its accumulation in the nucleus and its subsequent transcriptional activation in favor of the proapoptotic protein BAX target gene in the concomitant downregulation of its dominant negative, the Bcl-2 protein [100], leading to an increase in mitochondrial permeability, the cytosolic release of cytochrome c, and the execution of apoptosis [99,113,114,115]. In addition, it has been reported that exposure for 48 h to CAP (100 µM) downregulates the expression of Mdm2 in human colorectal HCT 116 cancer cells, supporting an amplification of the transduced apoptotic signal [116]. On the other hand, in this work, it was observed that p53 knockout isogenic clones presented a significantly higher tolerance when exposed to the same treatment, suggesting p53 dependence on the apoptotic effects of CAP. This was also reported in gastric cancer cells (AGS) that, after being depleted of p53 via siRNA methodology, experienced abrogation in the proteolytic activation of caspases 9 and 3, preserving the mitochondrial residence of cytochrome c and survival after an incubation of 12 h to CAP (200 µM), while their p53 wild-type counterparts underwent a significant increase in apoptotic activity [115].

### 4.1. Contribution of Oxidative Stress in the Proapoptotic Activity of Capsaicin in Cancer

The dissipation of the mitochondrial transmembrane potential (ΔΨm) in cancer cells exposed to CAP constitutes another of the events widely reported in the literature [98,117,118,119]. Although this metabolic alteration is constitutive in all apoptotic processes, the experimental demonstration of the influence of CAP agonism on TRPV1 on the ΔΨm dissipation, and, thus, in its cytotoxic effects on certain tumoral types, suggests that the mitochondrial oxidative stress produced (typically after 2–4 h of CAP exposure) could coexist and act synergistically with that of extramitochondrial origin, derived from the tNOX abrogation mediated by CAP. Given that, this second source is privative of cancer cells [82,83], and the oxidative stress threshold reached by these could be potentially higher than that caused in normal cells (tNOX^−^) when exposed to CAP, possibly also contributing to the selectivity of its effects. In this sense, the collapse of the ΔΨm induced by CAP could constitute a direct effect of its agonism on TRPV1 and the consequent increase in cytosolic concentrations of Ca^2+^, whose escalation at the level of mitochondria could cause their dysfunction and determine an increase in the production of reactive oxygen species (ROS), the opening of the mitochondrial permeability transition pore (mPTP), and the execution of intrinsic apoptosis, as reported in cells of two varieties of anaplastic thyroid cancer (8505C and FRO cell lines), papillary thyroid carcinoma (BCPAP) and follicular thyroid carcinoma cells (FTC-133), exposed to CAP over 24 h (50–200 µΜ). It should be noted that in this work, thyroid epithelial cells (Nthy-ori-3.1) presented a significantly higher tolerance to CAP in terms of their IC_50_ (probably due to their condition tNOX^−^). Finally, in this study, the pharmacological inhibition of TRPV1 (capsazepine) led to the mitigation of Ca^2+^ influx, lower oxidative stress, and a drop in CAP cytotoxicity, the same as the intracellular Ca^2+^ chelation (BAPTA tetrapotassium salt), confirming the participation of both the ion channel and the Ca^2+^ concentration in these cell types [117].

In what is possibly a redundant mechanism of the capsaicinoid-induced dissipation of ΔΨm, both CAP and DHC have been reported to operate as coenzyme Q antagonists through competitive binding to the ubiquinone-binding site at the level of complexes I (NADH-ubiquinone oxidoreductase) and III (ubiquinolcytochrome c oxidoreductase) [120,121] of the electron transport chain, with the subsequent generation of mitochondrial oxidative stress. This was demonstrated in cells of two varieties of human pancreatic adenocarcinoma (AsPC-1 and BxPC-3) previously exposed to CAP (150 µM). Pramanik et al. (2011) observed that the treatment of both tumoral types led to rapid and significant increases in superoxide (O_2_^−^) and hydrogen peroxide (H_2_O_2_) generation, whose maximum level was reached after only 2 h of exposure, to later be succeeded by the abolition of the activity of the antioxidant enzymes superoxide dismutase (SOD) and catalase. After 20 h, ~2.7- and 4.0-fold increases in cell apoptosis were, respectively observed. Detailed analyses of mitochondria isolated from both cell types revealed that CAP treatment caused drops in the activity of complexes I and III (significant from the 4th hour of exposure) in correlation with a decrease in its abundance [119]. Remarkably, in this same work, cells derived from the normal epithelium of human pancreatic ducts (HPDE6-E6E7) showed high tolerance to the same treatment by keeping their ROS levels stable during the 24 h of exposure, the same as their levels of apoptosis, despite having also registered a drop in the activity of complex III. Again, this could be associated with the absence of extramitochondrial oxidative stress derived from tNOX abrogation, which, unlike the two tumor types, allowed the HPDE6-E6E7 cell system to re-establish its redox homeostasis by endogenous antioxidant mechanisms without ceasing to experience some level of affectation at the level of complex III mediated by CAP. Interestingly, Wang et al. (2015) also explored the response of HPDE6-E6E7 cells to CAP, and found that their oncogenic activation through K-ras (G12V) transfection led to tNOX activation in this cell system, which caused marked increases in the level of oxidative stress and the apoptotic response secondary to CAP treatment at concentrations of only 5 and 10 μM during an unusually long exposure period (14 days), without observing similar effects in the HPDE6-E6E7 parental cells (tNOX^−^) [39].

Finally, supporting the notion of a synergy between oxidative stress caused by tNOX abrogation and that produced at the mitochondrial level, epithelial cells derived from human mammary fibrocystic gland (MCF 10A) transfected with a plasmid for the expression of tNOX also experienced a significant increase in their apoptotic responsiveness to CAP compared to their wild-type tNOX^−^ isogenic clones [92]. Therefore, although oxidative stress of mitochondrial origin (either with or without TRPV1 involvement) seems to constitute a preponderant mechanistic axis in the response of certain types of tumors to CAP, the relevance of oxidative stress of extramitochondrial origin (derived from the derogation of tNOX) could be proportionally greater, since the introduction of the “tNOX” variable seems to ensure a resolution by apoptosis even in noncancerous cell types.

### 4.2. Extrinsic Apoptosis Events Involved in the Cytotoxic Activity of Capsaicin in Cancer

On the other hand, some data have proven the involvement of events of the extrinsic pathway of apoptosis in the effects of CAP on certain cancer cell types. In urothelial papilloma cells (RT4 line) exposed to a concentration of 100 µM, the activation of the serine/threonine kinase ataxia telangiectasia-mutated (ATM) was detected, whose known role in the response to DNA damage due to oxidative stress [122] allowed its involvement to be predictable. After 1–3 h, ATM supported the phosphorylation of p53 at the level of its Ser15 residue (also reported in NB4 leukemia cells exposed to CAP 50 µM [123]) and shortly after (6–12 h) at Ser20 and 392, supporting an upregulation in the transcription and translation of the Fas/CD95 death receptor, which, after redistribution through the plasma membrane to a group with TRPV1 receptors, led to the activation of caspases 8 and 9, as well as truncated BID (tBID/p15). Interestingly, the Fas/CD95-TRPV1 cluster caused apoptotic signal transduction from the cell surface in the absence of the Fas/CD95 ligand (FasL), which was not detected in the treated cultures [124]. In this regard, it has been reported that in the context of nociception, the N-terminus of the TRPV1 receptor can bind to the FAS-associated factor-1 (FAF1) [125] protein that, in certain models, operates as an adapter of the Fas/CD95 receptor in the conformation of the denominated FAS death-inducing signaling complex [126], which could suggest its possible participation as a scaffold for the formation of the TRPV1-Fas/CD95 cluster. On the contrary, in another study, FAF1 degradation was directly postulated as a mediating event of CAP-induced apoptosis in two cell types of murine fibrosarcoma induced by methylcholanthrene (Meth A and CMS5 cells), whose exposure to CAP (100 µM for 72 h) led to rapid increases in ROS generation (also starting from 2 h, with a maximum peak during the 4th h), FAF1 degradation, and significant increases in cell apoptosis. To support their postulate, the authors induced FAF1 depletion using siRNA methodology, which improved CAP cytotoxicity ∼5-fold in both tumor types [41]. This apparent pleiotropy in the role of FAF1 during the induction of extrinsic apoptosis reveals the need to delve into the possible supramolecular interactions that could explain its role as a promoter or suppressor scaffold for cell death in response to CAP treatment.

On the other hand, a similar association has been reported between a subtoxic scheme of CAP exposure (50 µM during 30 min) and increases in the expression of death receptor 5 (DR5), which caused cells from five different varieties of human glioblastoma (SNU-444, U-87 MG, and U343, including two glioblastomas multiforme, T98G and U-251MG) to be significantly more susceptible to tumor necrosis factor-related apoptosis-inducing ligands (TRAIL; >25 ng/mL over 16 h). Treatment with CAP and TRAIL inhibited the ability of Cyclin-dependent kinase 2 (Cdc2) to phosphorylate Thr34 of the apoptosis inhibitor survivin, compromising its stabilization and leading to proteasomal degradation. In this sense, non-transformed human astrocytes from a primary culture showed greater tolerance to CAP, as they did not experience any of the mentioned effects when exposed to the same treatment [37].

Other reports reproduce the events described so far in a single experimental model, suggesting their joint action in favor of apoptotic execution. Kim et al. (2009) [116] showed that after an exposure of 24 h to CAP (100 µM), human colorectal cancer cells (HCT 116) experienced stabilization and the transcriptional activation of p53, with the concomitant activation of Fas/CD95 and DR4 receptors. Additionally, the combined treatment of CAP (100 µM) and the phenolic compound resveratrol (50 µM) substantially ameliorated these effects, from 6 h of exposure and up to 48 h. Figure 2 briefly outlines the major molecular events underlying the apoptosis mediated by CAP on cancer cells.

Finally, in light of these results, it has been proposed that a therapeutic strategy combining vanilloids (such as CAP) and chemotherapeutic drugs could be viable [127]. In this sense, improvements in the in vitro chemotherapeutic capacities of cisplatin and 5-fluorouracil have been reported, with the joint exposition of CAP (>50 μM for 24 h) in experimental treatments in gastric signet-ring-cell adenocarcinoma derived from metastatic site (SNU-668 line) and in gastric carcinoma also derived from metastatic sites (HGC-27 cells treated with CAP 0.3 mM for 24 h), which could constitute a first approximation to the proof of concept of CAP as a chemosensitizing agent against chemoresistant tumor cells [128,129].

### 4.3. In Vivo Antitumor Activity of Capsaicin and Expectations for Its Clinical Evaluation

On the other hand, the evaluation of CAP’s anticancer potential also includes a variety of in vivo trials, whose results adequately validate some mechanistic notions derived from in vitro studies. Such reports refer mainly to rodent models challenged with tumor xenografts (human cancers) or allografts (rodent tumor tissue) and, to a lesser extent, some trials with models of chemically induced carcinogenesis. In all cases, four routes of CAP administration have been tested with different levels of antitumor efficacy: oral (added to a standard diet), intragastric gavage (i.g.), and intraperitoneal (i.p.) and intratumoral injection [42,123,130,131,132,133,134]. Since the results are relatively homogeneous, here we will only address some representative works, reserving the discussion of the immunological effects observed in animal models of cancer treated with CAP for Section 7.

Lau et al. (2014) reported that CAP supported a significant decrease in the growth rate of tumor xenografts of human small cell lung cancer cells (DMS 53) implanted into athymic nude mice fed a standard diet + 10 mg CAP/kg body weight. By evaluating caspase-3 activity, immunoassays for cell death, and cleaved PARP determinations, a ∼3-fold higher apoptotic activity was demonstrated in tumor lysates from CAP-treated animals compared to vehicle [38]. Similar results have also been reported in athymic nude mice implanted with tumor xenografts of human pancreatic cancer cells derived from the primary tumor and metastatic site (AsPC-1 and BxPC-3 cells, respectively) treated with CAP concentrations as low as 2.5 or 5 mg/kg of body weight i.g. 5 days a week and a special antioxidant-free diet [39,98,135]. Additional analyses on tumor specimens from the CAP-treated rodents confirmed the elevation of oxidative damage parameters [39], as well as the loss of the mitochondrial permeability of cancer cells in the mass [135]. The intraperitoneal administration of CAP has also been tested at doses up to 10*x* higher (50 mg/kg) in immunodeficient mice (NOD/SCID) with established xenografts of human myeloid leukemia cells (NB4), reproducing the same antitumor response without evidence of coexisting toxicity [123].

In parallel, some studies with chemically induced carcinogenesis models in rodents have been developed, achieving consistent results. In their series of works on Swiss albino mice, Anandakumar et al. (2008, 2009a, 2009b, 2012, 2013) reported that the i.p. administration of CAP dissolved in olive oil (10 mg/kg, starting one week before the initiation phase and continuing for a further 14 weeks) was effective in restricting benzopyrene-induced (50 mg/kg bodyweight, also dissolved in olive oil) lung carcinogenesis by promoting apoptosis in the tumor mass. Likewise, CAP treatment reversed alterations in glucose metabolism and mitigated lysosomal abnormalities in lung tissue, which were concurrent events in this carcinogenesis model [136,137,138,139,140]. Similarly, but with a chemopreventive approach, Yoshitam et al. (2001) [141] evaluated the effects of the dietary addition of CAP (500 ppm) at four and twelve weeks on the initiation phase of chemically induced tumorigenesis by azoxymethane (AOM, two-weekly subcutaneous administrations of 20 mg/kg body weight) in Fischer 344 rats. The CAP addition led to significant reductions of 56 and 39% (*p* < 0.001 bis) in the frequency of aberrant crypt foci (ACF) per animal, in weeks four and twelve of the trial, respectively. Notably, the treatment was also effective in significantly reducing the occurrence of lesions with further progression, with 55% fewer colonic adenocarcinomas having been observed (*p* = 0.0450), demonstrating an appreciable chemopreventive activity on the progression from preneoplasia to malignancy. Finally, in this same work, but under a subacute scheme, the i.g. administration of CAP (40, 200, and 400 mg/kg of body weight/day) for five days led to dose-dependent elevations in the activity of the phase II enzymes glutathione S-transferase (*p* < 0.001) and quinone reductase (*p* < 0.005), both in liver and in large intestine. This was suggested by the authors as the underlying cause of the chemopreventive activity of CAP (given the role of phase I and II enzymes in AOM biotransformation) rather than an induction of apoptosis in the tumor mass, since no statistically significant differences were observed in the apoptotic index at 12 weeks in the lesions of the treated animals compared to the control group [141].

Based on this body of evidence, further investigations of the antitumor potential of CAP at the clinical level could be as pertinent as is plausible. In this sense, and under the parameters of the human equivalent dose (HED) [142], the administration of the lowest dose of CAP proven to achieve antitumor effects in mice (2.5 mg CAP/kg orally) [119] could suppose a starting dose for pharmacokinetic and bioavailability trials of CAP in humans of 0.202 mg/kg, which would be equivalent to administering ∼14.14 mg of CAP to a 70 kg human subject. However, at present, clinical trials addressing this gap are, to the best of our knowledge, non-existent. It has been suggested that the dissonance implied by the likewise postulated pro-tumoral effects of capsaicinoids (which will be addressed in the following sections), together with the limitations in terms of the pungency, hydrophobicity, low stability, and poor bioavailability of CAP, constitute some of the greatest deterrents to conducting these clinical trials [1].

In this sense, novel targeted delivery methodologies could help to overcome limitations of a physicochemical nature. Methods such as the encapsulation of CAP within the internal aqueous phase of double emulsions of “water-in-oil-in-water” (W/O/W) represent a promising resource for the enteric release of CAP by overcoming aspects such as its low solubility in water and high irritation potential, as observed in mouse gastrointestinal tissues. In the emulsion reported by He et al. (2023) [143], ethanol was used as the CAP solvent, whose mixture with pectin led to the formation of pectin hydrogels loaded with the capsaicinoid in what constituted the internal aqueous phase. The resulting double emulsion presented adequate stability, as well as a high encapsulation efficiency, whose performance in simulated digestion models at mouth and stomach levels confirmed the preservation of the compartmentalized integrity of the double emulsion, focusing its digestion and CAP release in the small intestine, which significantly improved its bioavailability—an aspect associated by the authors with the formation of mixed micelles from the digested lipid phase [143].

## 5. An In-Depth Molecular Perspective on the “Double-Edged Sword” Postulate

Despite the growing mechanistic understanding of the antitumor properties of CAP, the available data reveal that prolonged exposure to low concentrations (0.1 to 10 µM) leads to the development of aggressive phenotypes in tumor cells by increasing tNOX expression and ERK 1/2 phosphorylation in a concentration-dependent manner. These findings represent a line of evidence that could support the postulates of a series of papers that, 28 years ago [45,144], referred to the role of capsaicinoids as co-carcinogen agents, based on a series of in vivo data [44], mainly by testing extracts prepared from various matrices of *Capsicum* fruits and a whole miscellany of solvent systems.

Under the order proposed above, Liu et al. (2012) monitored the electrical impedance caused by cells of human colorectal carcinoma (HCT 116) cultured for 80 h in the presence of CAP concentrations between 0.1 and 10 µM. Their findings included a concentration-dependent increase in the proliferation and migratory capacity of tumor cells in Boyden’s chamber assay, regarding the group treated with vehicle [86]. Similar effects on cell proliferation were later reported in the cells of well-differentiated human transitional cell carcinoma of the urinary bladder (TSGH-8301 cells) exposed for only 23 h to 10 µM CAP [94], and, recently, by another research group in cells derived from human malignant melanoma (A375) exposed for 72 h to 10 µM CAP [134].

Further analysis by Liu et al. (2012) [86] revealed that after 24 h of exposure, treated HCT 116 cells acquired a mesenchymal phenotype to the detriment of their epithelial characteristics, in an in vitro representation of the so-called epithelial–mesenchymal transition (EMT), a latent embryonic program whose aberrant activation in cancer cells is associated with greater metastatic competence [145]. In this regard, a pronounced reduction in the expression of the constitutive proteins in the intercellular junction complexes E-cadherin and Zonula Occludens-1 (ZO-1) was observed after the treatment. There is a reported capacity of these proteins to, respectively sequester β-catenin and ZO-1-associated nucleic acid binding protein (ZONAB) in intercellular junction complexes, thus disrupting their import into the nucleus to the detriment of their transcriptional activity [146,147]. The downregulation of E-cadherin and ZO-1 could result in greater freedom for the transcription factors β-catenin and ZONAB to operate in favor of cell proliferation, as was observed.

In addition, HCT 116 cells increased in a concentration-dependent manner the expression of the mesenchymal markers N-cadherin and vimentin. It should be noted that a greater availability of N-cadherin is more than a parameter for the EMT evaluation, as it actively contributes to tumor progression and invasiveness. It has recently been reported that after inducing the expression of the soluble N-cadherin cytoplasmic domain, canine kidney cells (MDCK) experienced increased proliferation and growth in multiple layers due to the loss of contact inhibition. Likewise, some cells detached from the monolayer and evaded anoikis in correlation with mitogen-activated protein kinase/ERK phosphorylation (phospho-MAPK/ERK), supporting their survival and proliferation in the absence of extracellular matrix (ECM) anchorage [148], which is both a hallmark of malignant transformation and a characteristic that enables transformed cells to remain viable during their circulatory dissemination [149].

Detailing the contribution of phospho-MAPK/ERK in the evasion of anoikis, it is conventionally accepted as a consequence of the oncogenic signaling of receptors with tyrosine kinase activity epidermal growth factor receptor (EGFR) and Erb-B2 Receptor 2 (ErbB2) [150,151,152]. In this context, recent data allow a better understanding of downstream events in phospho-MAPK/ERK signaling. Using the inherent properties that allow inflammatory breast cancer (IBC) cells to proliferate in suspension in lymph and not only in solid tumors, Buchheit et al. (2015) [153] demonstrated that the phosphorylation (Ser59) of the proapoptotic protein BIM-extra long (Bim-EL) by phospho-MAPK/ERK is a key event for survival in the absence of the anchorage of IBC cells (SUM149 and KPL-4) due to a concomitant block in the activation of caspases. In this regard, endogenous protein co-immunoprecipitation assays revealed that Bim-EL phosphorylation led to its hetero-trimerization with LC8 and Beclin-1 proteins, thus limiting the subcellular localization of Bim-EL away from the mitochondria (to the detriment of its antagonistic activity on the antiapoptotic proteins of the Bcl-2 family), favoring the opening of mPTP, the activation of effector caspases, and the subsequent execution of anoikis. In this sense, the pharmacological inhibition of phospho-MAPK/ERK effectively abrogated the coprecipitation of Bim-EL and LC8 with Beclin-1, enhancing the immunofluorescent colocalization of Bim-EL and mitochondria in favor of resolution by anoikis. However, in other cell types, Bim-EL Ser69 phosphorylation leads to its ubiquitylation and subsequent proteasomal degradation [154], so the variety of mechanisms underlying cell survival in an anchorage-independent manner include a number of events with convergence in the phosphorylation of Bim-EL, making it possible to foresee their involvement in the cancer cell behaviors described by Liu et al. (2012) [86] in response to CAP.

### 5.1. Modulation of Intracellular Calcium and Its Possible Involvement in the Promoting Effects of Capsaicin at Low Concentrations

CAP’s cancer-promoting properties can also theoretically be associated with its agonism on TRPV1 and subsequent Ca^2+^ internalization, effectively induced at CAP concentrations as low as ≤20 µM [155]. This cation, operating as a second messenger, constitutes a positive regulator of cell migration; it is particularly effective in cancer cells, apparently due to its aberrant expression of Ca^2+^-handling proteins and Ca^2+^-dependent effectors, either in the form of kinases, proteases, or phosphatases. These constituents collectively enable the cell to exhibit a more rapid FA turnover, as well as the more efficient proteolysis of extracellular matrix components [156]. In this sense, it has been shown that the availability of Ca^2+^ plays a determining role in the development of an invasive phenotype in breast cancer cells with pleural metastasis (MDA-MB-468), which, after having been subjected to the intracellular chelation of Ca^2+^, experienced a pronounced inhibition in the phosphorylation of STAT3, restricting the EMT induced by epidermal growth factor (EGF) by significantly repressing the transcriptional expression of mesenchymal markers such as N-cadherin, Twist, and the CD44/CD24 ratio, and at a translational level, to the vimentin protein. Although poorly understood, the mechanisms underlying such responses were partially dependent on the availability of the transient receptor potential melastatin-like 7 (TRPM7) channels, whose depletion resulted in the cessation of vimentin expression, as well as STAT3 phosphorylation; thus, the involvement of other Ca^2+^ transporters such as TRPV1 could be plausible [157]. However, it is clear that for a given cancer cell system to take advantage of an increase in intracellular Ca^2+^ concentration, secondary to CAP agonism over TRPV1, its magnitude should not reach a level that implies mitochondrial dysfunction, a factor that, for its part, it could also be involved in the pleiotropy of the outcome observed with higher concentrations of CAP.

On the other hand, the ligand-mediated physiological activity of TRPV1 is subject to the molecular countermeasures executed by CaM and PPP3C in the cellular environment. Therefore, although there is still a need to experimentally verify the involvement of intracellular Ca^2+^ in the promoter effects produced by low concentrations of CAP in cancer cells, this model could also be a suitable platform for the study of possible oncogenic dysfunction in the regulatory roles of CaM and PPP3C on the TRPV1 ion channel. Figure 3 proposes a mechanistic model for the promoter effects of CAP at low concentrations on cancer cells.

### 5.2. Promoter Effects of Capsicum Fruits Extracts in Animal Models of Chemically Induced and Spontaneous Tumorigenesis

On the other hand, the cancer-promoting potential of capsaicinoids has been addressed in a series of epidemiological studies. Various authors have proposed a positive correlation between chili consumption and the development of certain malignancies in human populations, principally stomach and liver cancers, e.g., in North American ethnic minorities (Mexican American) who consume chili [158]. Later, an association between the Mexican population and the risk of developing gastric cancer was reported by López-Carrillo et al. (2012) [159], who also considered other risk factors in their analysis. In this sense, the meta-analysis by Pabalan et al. (2014) stands out for having integrated 10 studies with a case–control design with gastric cancer as the outcome, for a total of 2452 cases per 3996 controls. The results of their interaction tests (95% confidence interval) suggested a protective effect (O.R., 0.55; *p* = 0.003) at low intakes of CAP, while a greater susceptibility (O.R., 1.94; *p* = 0, 0004) at medium–high intakes, preserving the heterogeneity condition (*P*_heterogeneity_ ≤ 0.0001) [160]. Certain experimental reports with rodent models are recurrently referred as antecedents in these epidemiological studies, such as the classic work of Agrawal et al. (1986), who reported promoting cancer effects by the administration of an ethanolic extract of chili, with an estimated CAP concentration of 2.5 mg/mL together with other natural compounds (coexisting contaminants [161] with affinity for the solvent used during extraction [44]) in N-nitroso-acetoxymethyl-methylamine (DMN-)-induced stomach and liver carcinogenesis in albino mice [162]. Similar results were reported with the initiating agent N-methyl-N-nitro-N-nitrosoguanidine (MNNG) in the stomach of Fisher rats fed with 1 and 3% of red-hot pepper on their diet over 37 weeks. These groups presented an incidence of tumors of 57 and 63%, respectively, exceeding the 44 cases observed in the control group treated with MNNG only [163]; nevertheless, in these works, no evidence of tumorigenesis was found in the groups treated only with chili.

The treatments used consisted of complex mixtures with an appreciable (although very probably variable) concentration of capsaicinoids and other constituents of the *Capsicum* fruit, given the limitations in the extraction methodologies and derived from the sources of plant material [44], so their findings could be subject to substantial variability. In this sense, other authors have indicated both harmless and chemopreventive effects in natural mixtures of capsaicinoids (~64.5% CAP, ~32.6% DHC) powdered in the food pellet. Using a chronic exposition scheme (79 weeks) in B6C3F1 mice, Akagi et al. (1998) did not find an increase in the incidence of tumors. On the contrary, they observed a negative correlation between the incidence of bearing neoplasms in females and the provided doses of capsaicinoids, the same as the occurrence of hepatocellular tumors in both sexes [164].

Transferring the approach to the human case, to the best of our knowledge, studies on the effects of long-term chili consumption in human patients with active cancer remain undeveloped. However, it is clear that a case–control design addressing this question will not fill the gap of a randomized clinical trial that uses a suitable pharmaceutical form of CAP at an appropriate dosage.

### 5.3. Cancer-Promoting Effects at Target Concentrations of Capsaicin: A Potential Risk?

Returning to the line of evidence involving pure CAP, the contributions of Caprodossi et al. (2011) on the exacerbation of aggressive phenotypes induced by the capsaicinoid considerably expand the resolution of the model. Their results also stand out for having been produced at unusually high concentrations of the capsaicinoid (100 µM). In TRPV1^−^ human primary bladder carcinoma cells (5637 cell line), whose phenotype was associated with persistent stability both in intracellular Ca^2+^ concentrations (evaluated through FLUO 3-AM fluorescence, in an observation range 0–120 s) and apoptotic activity, did not register significant changes after 24 h of exposure to CAP [165]. Unlike the observations of Liu et al. (2012) [86] and Lin et al. (2016) [94], in this study, no changes were observed in the proliferation of cells exposed to a range of 25 to 100 µM of CAP (24 h); instead, a significant improvement was observed in their metastatic competencies in the Matrigel invasion assay (CAP 100 µM for 24 h). Likewise, increases in the transcription of insulin-like growth factor-1 (IGF-1) and its receptor (IGF-1R) were detected, as well as IGF-1 levels in the supernatant of the treated cultures, with statistical significance from the 6th hour of exposure, and these levels remained on the rise until 24 h.

Also, in the context of 5637 cell migration and invasiveness, it has been shown that the activation of the MAPK/ERK pathway constitutes an important downstream objective in IGF-1R signaling by positively modulating the activity of the heterodimeric transcription factor Activator Protein 1 (AP-1). Significant increases (~5-fold) in levels of one of AP-1′s transcriptional products, matrix metalloprotease-9 (MMP-9) [166], were detected both at the level of gene expression and its post-translational activation after 24 h of CAP 100 µM treatment. Although the proteolysis of the extracellular matrix mediated by MMP-9 culminates in promoter events induced by subtoxic concentrations of the capsaicinoid in certain tumor types, Caprodossi et al. (2011) also reported increases in the expression of three proangiogenic genes (vascular endothelial growth factor, angiopoietin 1 and 2), six prometastatic genes in addition to MMP-9, and surprisingly, a downregulation in the expression of Fas/CD95 and tumor necrosis factor receptor 1 (TNFR1) [165].

Finally, the atypical aggressive behavior of 5637 cells exposed to CAP concentrations in the order of magnitude of two was completely abolished after forcing the expression of TRPV1 (through transfection of its cDNA coding sequence) into the 5637 cellular system (TRPV1^+^-5637). The cells radically replaced the behavior of what we might call a “typical response” of cancer cells exposed to CAP in terms of the sudden increase in the influx of intracellular Ca^2+^, affectations in cell proliferation (from 50 µM, *p* ≤ 0.01), and significant increases in cell apoptosis (~10-fold compared to vehicle) after 24 *h* exposure. Although the analyses carried out by Caprodossi et al. (2011) [165] did not include the evaluation of oxidative stress, it is reasonable to hypothesize its involvement in the effects observed in TRPV1^+^-5637 cells treated with CAP, as well as the dissipation of ΔΨm.

Notwithstanding the abovementioned studies, it is clear that the hypothesis about the possible molecular attributes of cancer that could confer the ability to develop EMT secondary to its exposure to CAP in typically effective concentrations to achieve antitumor effects cannot be resolved from the data obtained from the only tumor type reported to develop these behaviors (5637 cells). In particular, the same cell type has been described by Chen et al. (2012) and Zheng et al. (2016) as being responsive to CAP concentrations ≤50 µM for 48 h in terms of a significant loss in cell viability (*p* < 0.01), cell cycle arrest at the G0/G1 phase, and loss of mitochondrial permeability [167,168]. Although these differences from the findings around the model of exposure of cell type 5637 to CAP (50–200 µM) could be reconciled as “early” (0–24 h) and “late” events (24–48 h) in cytotoxicity kinetics, it should be noted that these late authors also reported that the 5637-cell system exhibits a high expression of TRPV1, in contrast to the findings described by Caprodossi et al. (2011) [165]. Other factors, such as the expression level of tNOX and the probable absence of TRPV6 in 5637 and TMC-1 [83,95] cells, could have scope as variables both in terms of the study of the mechanisms that underlie their refractoriness to CAP and in terms of the possible identification of tumor phenotypes that could indicate signs of contraindication to experimental therapy with capsaicinoids.

On the other hand, it is interesting that stimulatory effects on tNOX expression and consequent increases on proliferation and invasiveness have been reported in lung cancer cells (line A549) when they were exposed to low concentrations of the chemotherapeutic agents doxorubicin and tamoxifen [169]. Since doxorubicin (an anthracycline [170]), tamoxifen (selective estrogen receptor modulator [171]), and CAP act through considerably different mechanisms, it seems reasonable to speculate that the findings of Liu et al. (2012) [86] and Su et al. (2012) [169] constitute, as a whole, a common “countermeasure” of malignant cells after prolonged exposure to non-lethal concentrations of cytotoxic compounds. In this sense, and in contrast to the promoting effects observed after prolonged exposure to low CAP concentrations, 100 and 250 µM were effective for downregulating tNOX, producing inhibitory effects and apoptosis both in HCT 116 [116,172] and TSGH-8301 cells [94].

Additionally, it is worth mentioning that the outcomes, as well the biochemical-mechanistic principles abstracted from models of exposure to the CAP concentration range of 0.1–10 µM, are subject to remarkable variability among different cancers in vitro, e.g., cells of the LNCaP, PC-3, and DU-145 lines (prostate cancer), MCF7 (breast adenocarcinoma), T24 (transitional cell bladder cancer), and a cellular model tNOX^+^, built from the normal epithelium of human pancreatic ducts (HPDE6-E6E7) with oncogenic activation through K-ras (G12V) transfection, experience CAP-mediated increases in oxidative stress and inhibitory effects from concentrations as low as 10 µM at exposure times between 24 and 48 h [39,94,173,174]. Moreover, experimental concentrations between 50 and 250 µM lead, with few exceptions, to generalized apoptotic effects in cancer cell cultures [99,175]. Collectively, these data demonstrate that the outcome of cancer cell exposure to CAP is defined in terms of its concentration, in association with the genetic background of a given cancer type, being especially relevant for the expression levels of TRPV6 and tNOX (as possible predictors of responsiveness) as well the phenotype TRPV1^High^ (as a possible eligibility criterion). In this regard, the typification by markers of cancers potentially responsive to CAP could constitute a fundamental aspect in the delimitation of a dose with antitumor efficacy with a view to its clinical evaluation in human subjects.

## 6. Capsaicinoids as Autophagy Inducers in Cancer and Non-Cancer Cells

Although the capsaicinoid family is made up of 22 analogous compounds [2] and 5 of them are widely referred to in the literature [12,48], the information available on their biological effects is almost exclusively limited to CAP [176]. Much less studied are the effects of DHC on cellular systems in vitro, which been associated with the induction of autophagic death mediated by catalase [177,178,179]. Although these data are limited to the reports of one research group, their results in a variety of cell lines, including both cancer and noncancerous cells, have shown notable differences with respect to the in vitro cytotoxic effects typically observed with CAP.

After 24 h of DHC exposure (200 µM), mammary adenocarcinoma and colon cancer cells (MCF7 and HCT 116, respectively) underwent autophagy death at slightly higher rates than those observed for CAP at the same concentrations; moreover, CAP, for its part, was ineffective in inducing apoptosis under the study’s experimental conditions [177]. Interestingly, the autophagy induced by DHC was mediated by the activation of catalase, whose expression increased after 30 min of exposure, suppressing ROS levels in a manner that positively correlated with phosphatidylethanolamine (PE) conjugation to the cytosolic form of the microtubule-associated protein 1A/1B light chain (ultimately LC3-II), an event that is essential for the formation of the autophagosome and the proper execution of mammalian autophagy [180]. Also, in this study, Seon et al. (2008) [177] found that after the pharmacological inhibition of catalase with 3-amino-1,2,4-triazole (3-AT; pretreatment with 25 mM per 2 h), there was a cessation of PE-LC3 conjugation, abrogating autophagy and favoring resolution by apoptosis, demonstrating that the DHC-mediated autophagic cell death in MCF7 and HCT 116 cells was catalase-dependent and, unlike CAP-mediated apoptosis, was sustained by an antioxidant response in the cellular system.

However, with the execution of autophagy, the possible outcomes are diverse since, in principle, as an adaptive response of the cell to the depletion of nutritional resources, autophagy may result in a protective effect by keeping cells alive during nutrient deprivation or under other sources of stress, either through its bulk degradation of cytoplasmic proteins or in the form of the selective degradation of damaged or toxic cytoplasmic organelles, with the consequent recycling of the released macromolecules; conversely, if the source of stress continues or excessively induces autophagy, this leads to cell death [181]. In this sense, Choi et al. (2010) stated that DHC-mediated autophagy evoked a survival effort in other cell lineages, since it conferred greater resistance against death to cells derived from lung epithelial fibroblasts (WI38) and of non-small cell lung cancer (H1299), whose magnitudes of apoptosis and necrosis were lower than those observed in cells derived from malignant pleural effusion (H460) and lung cancer (A549), whose capacity to execute autophagy was lower. Likewise, DHC-mediated autophagy in the two types of cells most competent for its execution showed a dependence on catalase, whose inhibition by 3TA pretreatment led to drops in autophagic vacuolization and increased susceptibility to apoptosis after DHC exposure (200 μM for 6 h) [179].

In these works, DHC-induced autophagy was consistently mediated by catalase, whose activation increased in response to treatment, attenuating the levels of ROS in each cellular system. In contrast, the apoptotic effects of CAP are categorically linked to the generation of oxidative stress, which could suggest a probable antagonism between the cytotoxic mechanisms they exhibit. However, other authors who included autophagy parameters in the analysis of their in vitro CAP-exposure assays similarly found an increase in autophagic activity in cells of various tumor types, as well as one derived from normal cells, suggesting that both the activation of catalase and the execution of autophagy could constitute facets of a rheostatic response against the oxidative stress and damage produced by the treatment, and not precisely a consequence of the capsaicinoid used. For example, in human malignant glioblastoma tumor-derived cells (U-251 MG), CAP caused losses in cell viability in a concentration-dependent manner, starting at 25 μM and 24 h of exposure, whereas at 100 μM (24 h), the treatment led to significant increases in both LC3-II conjugation and the abundance of beclin-1 and p62 (sequestosome 1) proteins. Beclin-1 plays an important role in complexes to generate phosphorylated-PE, which is of great importance in autophagic vacuolation [182], while p62 operates as a cargo receptor for ubiquitinated targets for autophagosomal degradation [183]. Thus, the confluence of the increases in the three parameters indicated a complete autophagic flux, whose coexistence with the elevation of the PUMA-α protein in the U-251 MG cells was indicative of concurrent increases in cell apoptosis post treatment. Further analysis demonstrated that the addition of 3-Methyladenine (3-MA; 5mM), a pharmacological inhibitor of the autophagosome maturation, during the exposure phase decreased the abundance of LC3-II, Beclin-1, and p62; at the same time, it led to the stabilization of p53, favoring the expression of its transcriptional product PUMA-α and a resolution by apoptosis, which significantly increased [184].

Similar results were reported by Granato et al. (2015) in human primary effusion lymphoma cells (BCBL-1) exposed to CAP 200 μM per 24 h [185]. More recently, the same response was observed in tongue squamous cell carcinoma derived from a metastatic site on a cervical lymph node (HSC-3) with equal treatment [96], and in allografts of cells derived from mouse melanoma (B16F10 versus female C57BL/6 mice) after an intratumoral injection of 200 μg of CAP in PBS vehicle with 1.12% ethanol [134]. Noticeably, two parallel assays in these later works using cells of human tongue cancer (SAS) and derived from human malignant melanoma (A 375) revealed decreases in viability, exclusively at the expense of cellular autophagy and tNOX abrogation [96,134]. For the rest, either because the pharmacological inhibition of the autophagy pathway (through co-treatment with 3-MA 5–10 μM, 25 μM of the lysosome inhibitor chloroquine for 1 h, or a pretreatment with 10–100 nM of bafilomycin-A1 for 1 h) improved the post CAP-treatment apoptotic activity (on BCBL-1, HSC-3, SAS, and A 375 cells) or because cell autophagy subsided during the first 12 h of exposure, to be succeeded by increases in cell apoptosis (on HSC-3 cells), all of these cell types reproduced effects that confirm the role of autophagy as a survival effort in tumor cells exposed to CAP or DHC (100–200 μM, between 16 and 24 h). However, it can be translated in its own modality of cell death, as was observed in A 375 and SAS cells.

Regarding the predictability of the outcome of autophagy mediated by capsaicinoids, the efficacy and consistency shown by 3-AT (an irreversible catalase inhibitor [186] and not strictly an autophagy inhibitor) in its experimental inhibition [177,179] could suggest that the eventual impossibility to execute an antioxidant response able to mitigate the oxidative stress threshold produced by the treatment in a given cell lineage is a factor that could potentially predict whether capsaicinoid-induced cellular autophagy will cause cell death and not survival. In this sense, the CAP-tNOX/SIRT1/p53 mechanistic axis could provide elements to hypothesize that the phenotype tNOX^−^ of certain cell lineages could have contributed to the lower apoptotic responsiveness to CAP/DHC observed in the reports by Seon et al. (2008) [177] and Choi et al. (2010) [179]. tNOX absence would also imply the deprivation of extramitochondrial oxidative stress resulting from its capsaicinoid-mediated inhibition, leaving the ROS produced after the inhibition of complexes I and III of the electron transport chain as the only source [120,121], and whose magnitude could be insufficient to cause cell death and, on the contrary, admit rheostasis through endogenous antioxidant mechanisms and cell autophagy (as has been observed [40,178]) that could ultimately re-establish conditions compatible with life in tNOX^−/low^ cells treated with capsaicinoids.

Subscribing to the previously developed hypothesis, human fetal lung epithelial tNOX^−^ fibroblasts (MRC5 cells) not only did not reduce their NADH oxidation in the presence of CAP (200 μM for 24 h), but also increased the production of the oxidized species NAD^+^, which ultimately, operating as a coenzyme for SIRT1, catalyzed the higher execution of p53 deacetylation (Lys382)-dependent autophagy in these cells, without compromising their survival even after 108 h of exposure. This behavior was accompanied by significant increases (~3.4-fold at 24 h) in the abundance of the antiapoptotic protein Bcl-2, to the detriment of the proapoptotic BAX protein (−0.8-fold) [40]. Although oxidative stress was not evaluated in this work, these events could brace the notion of a re-establishment of redox homeostasis and a consequent reinforcing of survival pathways. Similarly, the group led by Seon and Choi, pioneers in the study of the biological effects of DHC, provided data obtained in other normal lung epithelial fibroblast cells (WI38) that could strengthen our hypothesis. However, while differences in responsiveness to DHC and CAP were also reported in this study (from concentrations > 200 μM per 24 h), the LC_50_ of both capsaicinoids were notably higher than those typically tested in cancer cells, being ~300 μM for DHC and greater than 400 μM for CAP, thanks to the remarkable autophagic competence of WI38 cells [178].

Although, for the moment, the available data are limited to two types of lung epithelial cells, the increases in NAD^+^ concentration in non-cancer cells exposed to CAP suggest that the constitutive NADH oxidase ENOX1 (or another wild-type protein of the ecto-NOX family) would act in the opposite direction to its aberrant counterpart tNOX when in contact with capsaicinoids in relation to its NAD^+^ production, which perhaps has some level of contribution to the higher tolerance exhibited by non-cancer cells, and probably implies a conditioning of the apoptotic response in tumors with a tNOX^−^ phenotype treated with CAP. However, experimental confirmation remains to be conducted, and this remains a major question related to capsaicinoid-mediated autophagy, along with the apparent differences in cytotoxic potencies displayed by CAP and DHC.

## 7. Immunomodulating Properties of Capsaicin in Cancer

Some studies involving in vivo and in vitro assays have shown that CAP is able to modulate certain facets of the immune response involved in immunosurveillance. This attractive property has earned CAP the status of lead compound, since its structure begins to inspire the synthesis of new molecules with the expectation of preserving the antitumor and immunomodulatory properties of CAP [187], while seeking to overcome the physicochemical limitations imposed by its molecular structure [1]. Kim et al. (2014) [47] found that pretreatment for 1 h with the capsaicinoid inhibited natural killer (NK) cell cytolytic activity in a concentration-dependent and TRPV1-independenent manner, starting at 10 μM (*p* < 0.05), with marked exacerbations at 50 and 100 μM (*p* < 0.01 and *p* < 0.001, respectively), both from NK isolated from the peripheral blood of healthy donors and of the NKL cell line (derived from NK cell lymphoblastic leukemia). The CAP pretreatment also reduced NK production of proinflammatory cytokines (interferon-γ and tumor necrosis factor-α; TNF-α). This lineage of effectors belonging to the innate immune system is considered the third most abundant lymphocyte population [188] and an essential component in the antitumor immune response [189]. Therefore, NK dysfunction induced by CAP could result in potentially dire consequences for immunosurveillance [47]. However, in this study, CAP was also effective in inducing apoptosis in gastric carcinoma (Hs746T), gastric adenocarcinoma in situ (AGS), and liver metastatic site-derived cells (MKN45), but higher concentrations (100 μM) and longer exposure times (24 h) than those required to cause disfunction in NK cells were required, which might suggest that the anticancer potential of CAP could coexist with undesirable effects on the function of NK cells.

Other authors have proposed that CAP could exert a positive influence on other cell lineages that also participate in the antitumor immune response. The first reports in this area described beneficial effects on the ex vivo maturation of murine (strain C57BL/6) dendritic cells (DCs) in terms of their expression of IA^b^ (mouse ortholog of major histocompatibility complex class II; MHC-II) and the costimulatory molecule CD86 (B7-2), after an in vitro treatment with CAP (to 50 and 100 µM) for 16 h, as well as increases in the migration of DCs towards draining lymph nodes in the rodents after the intradermal administration of 200 µg of CAP in 200 µL of vehicle (80% PBS, 10% ethanol, and 10% Tween 80) [190]. Subsequently, Granato et al. (2015) [185] studied the effects of CAP on two varieties of primary effusion lymphoma (PEL) cells (BC3 and BCBL1), whose production of anti-inflammatory cytokines and other soluble factors impairs the functional maturation of DCs, abolishing their ability to capture tumor antigens and perform their cross-presentation to CD4^+^ and CD8^+^ lymphocyte populations, thus preventing the differentiation of the latter into functional antitumor effectors [191]. Their results indicate that CAP (100 and 200 µM) induced apoptosis in PEL cells, linked to STAT3 dephosphorylation (Tyr705). The analysis of the apoptotic phenotype also revealed the presence of two damage-associated molecular patterns (DAMPs), CRT and heat shock protein 90 (HSP90), on the surface of the treated PEL cells [185]. Since this increased availability of CRT and HSP90 in the tumor microenvironment (TME) determines their interaction with a variety of immunological cell types (with a positive expression of cognate pattern recognition receptors; PRRs) capable of triggering a sterile inflammatory response and, potentially, to execute antigen-specific adaptive immune responses, their identity as DAMPs is well established [192], just like the CAP-mediated immunogenic cell death (ICD) of PEL cells. Interestingly, the authors found that CAP (150 µM per 24 h), by itself, supported the maturation of monocyte-derived DCs isolated from healthy donors. Equally notable was CAP’s ability to counteract the immunosuppressive effect of soluble factors released by PEL cells by restoring the differentiation of monocytes to DCs, in terms of their increase in the expression of CD1a and a CD14 decrease, after having been cultured for 5 days in proper differentiation medium with 20% supernatant of PEL cells with and without CAP (150 µM) [185].

The same working group studied the effects of the co-culture (2:1 ratio) of immature DCs and two varieties of human urinary bladder cancer cells (T24 and SD48) on a course of CAP-mediated ICD (150 μM per 30 and 12 h, respectively) [193]. Their results indicated that 48 h after starting the cocultivation, a substantial upregulation ensued in the differentiation markers CD83 and CD86 in DCs established from human peripheral blood-derived CD14^+^ monocytes in a CD91-dependent manner (a PRR common to HSP90, HSP70, and CRT). Again, CAP, by itself, significantly activated the DCs (starting at 50 µM), and the pharmacological inhibition of TRPV1 in them (through pretreatment with 10 mM capsazepine for 1 h) only partially affected its activation. However, 150 and 250 µM of CAP were cytotoxic for 25 and 50% of the immature DCs, respectively [194]. Taken together, these data consistently support the effects of CAP on DC activation, both by promoting it directly (by mechanisms only partially understood) and by inducing ICD in tumor cells, whose release of DAMPs might represent activation opportunities for the DCs in the TME (via PRRs), as well as for the phagocytosis of “tumor antigen-DAMP” clusters, which could lead to antigen cross-presentation and antigen-specific antitumor immune responses. However, the deleterious effects of CAP on the cells of the immune system (NK and DCs, according to current knowledge) could constitute a major challenge for the delimitation of a CAP concentration for human subjects and provide another reason to explore the performance of targeted delivery systems [143], as previously discussed in Section 4.3.

On the other hand, previous results reported by Beltran et al. (2007) [42], who studied the immunomodulatory activity of CAP in vivo against two varieties of syngeneic tumor allografts (the Meth-A line, derived from a fibrosarcoma chemically induced with 3-methylcholanthrene, and the CT-26 line, derived from undifferentiated colon carcinoma induced with N-nitroso-N-methyl urethane) ectopically established in BALB/cJ (+/+) and /cJ-*nu/nu* mice, demonstrated consistent effects with the concretion of an antitumor immune response secondary to two doses of intratumoral CAP of 100 or 200 μg (groups “*a*” and “*b*”, respectively), which might demonstrate good capsaicinoid tolerability in both DCs and cytotoxic T lymphocytes (CTL) in a direct route of administration scenario [42]. Their findings included generalized responses in terms of a slowdown in the growth rate of the injected tumors, even leading to complete tumor ablations in 40% of the treated animals in group “*a*” and 20% in group “*b*”. In a second trial of the same work, it was observed that after applying equal CAP treatment to an initial Meth-A tumor, there was a significant slowdown (*p* < 0.05) in the growth rate of a second non-injected Meth-A tumor in the same host compared to a corresponding control group with vehicle. In this same sense, fibrosarcoma tumors with a different antigenic signature (CMS-5 cells) did not experience secondary undermining of their growth rate after the CAP treatment of an initial Meth-A type tumor in the same individual. These observations demonstrated that the underlying cause of the effects observed in non-injected lesions was an antigen-specific antitumor immune response secondary to CAP treatment. Finally, the authors reported that Meth-A tumors established in *nu/nu* mice were not responsive to CAP treatment, which, in turn, showed that the antitumor immune response secondary to intralesional CAP treatment would be operated by CTLs [42].

Subsequently, and extending the description of the immunological events triggered by intralesional CAP in solid tumors, the same work group provided data demonstrating that two injections (100 μg) were not only effective for the complete ablation of poorly differentiated colorectal carcinoma allografts (CT-26 cells) ectopically established in BALB/cJ mice, but also to induce a potent CTL-mediated apoptotic response in tumor-associated stromal cells (CD11b^+^), since, unlike immunocompetent animals, BALBc-*nu/nu* mice subjected to the same treatment experienced the antitumor effect, but not the depletion of CD11b^+^ cells in the tumor stroma [133]. Completing the picture, a substantial remodeling of the TME was observed after CAP treatment in terms of increased concentrations of granulocyte–macrophage colony-stimulating factor (GM-CSF) and proinflammatory interleukin (IL)-12, as well as in a drop in the concentration of the anti-inflammatory IL-10. The establishment of this eminently proinflammatory milieu was succeeded by a depletion of lymphocytes with the immunosuppressive phenotype of regulatory T cells (T_Regs_; CD4^+^ CD25^+^ Foxp3^+^) and, ultimately, by increases in the presentation of tumor antigens by CD11b^+^ cells [133]. Described by the authors as a “sensitization of the tumor stroma”, this behavior could be indicative of a polarization of subpopulations of macrophages to the M1 phenotype, whose prolific secretory activity could also have played a key role in the recruitment and activation of NK, DCs, and CTLs [195]. Therefore, CAP’s immunostimulatory properties appear to outweigh its potential deleterious effects on immune cells by reactivating some of their behaviors against established cancer lesions. Thus, the antitumor action of capsaicinoids and the concretion of the immunological effects derived from them could potentially synergize the therapeutic response of immunological checkpoint inhibitors. However, this hypothesis remains unexplored.

## 8. Conclusions

At the dawn of a new stage in which the study of capsaicinoids in cancer has reached sufficiency for preclinical purposes, a large body of evidence concerning the effects of CAP and DHC in a plethora of cancer models places capsaicinoids in a position from which it is certainly possible to modulate signaling pathways with outcomes as disparate as cell death, survival, and even degeneration to more aggressive phenotypes of certain tumoral types. However, the pleiotropy of these effects seems to be related mostly to the dose (separated by a difference of at least two orders of magnitude in concentration), while the resistance of specific cell types to CAP reveals potentially useful variables, both for the study of the mechanisms that underlie its refractoriness and for the possible identification of tumor phenotypes that could indicate signs of contraindication to experimental therapy with capsaicinoids.

Molecular parameters such as the availability of TRPV6, the tNOX expression (as possible predictors of responsiveness), and the phenotype TRPV1^High^ (as a possible eligibility criterion) constitute a good starting rationale for the possible typification of those cancers that could respond to capsaicinoids therapy, an aspect that, together with the definition of an appropriate pharmaceutical form and the delimitation of a dosage with antitumor efficacy, could promote the evaluation of the anticancer potential of capsaicinoids at the clinical level, either as novel agents of category L01C “plant alkaloids and other natural products” or under the operational definition of the group L03AX for “Other immunostimulants” in terms of the Anatomical, Therapeutic, and Chemical classification system (ATC) [196].

## Figures and Tables

**Figure 2 cells-12-02573-f002:**
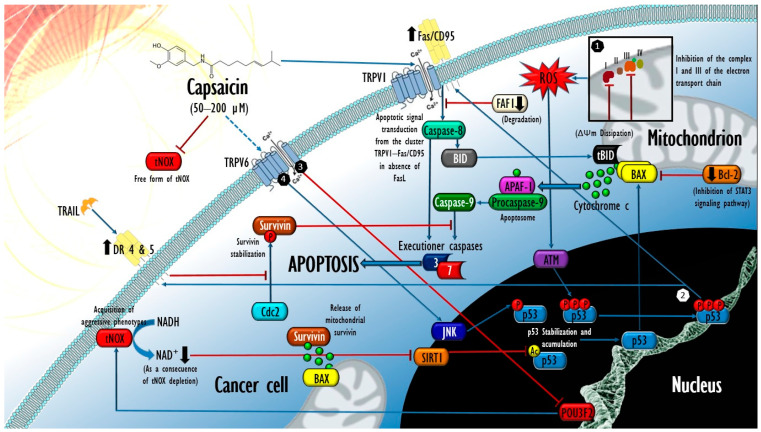
Main molecular events underlying CAP-mediated apoptosis in cancer cells. Intrinsic and extrinsic mechanisms mediated by p53. (1) After the inhibition of complexes I and III of the electron transport chain and the increase in ROS, the activation of ATM occurs, which phosphorylates p53 in three residues, leading to its transcriptional activation in favor of Fas/CD95, whose clustering with TRPV1 results in the transduction of a FasL-independent apoptotic signal on caspase-8 and tBID, followed by the cytosolic release of the mitochondrial proapoptotic factors cytochrome c and APAF-1, which, after recruiting procaspase-9 hetero-oligomerize (apoptosome) to activate caspase-9, supports the subsequent activation of the effectors caspase-3 and 7. (2) The transcriptional activity of p53 leads to increases in the expression of DR4 and DR5, which, in the presence of TRAIL, inhibit survivin stabilization, whereby the apoptotic signal is amplified. In a TRPV6-dependent manner, (3) CAP causes the depletion of POU3F2 and its transcriptional product tNOX, leading to a drop in NAD^+^ production that negatively affects SIRT1 deacetylase activity on p53, increasing its acetylation; in parallel, (4) JNK is enabled to phosphorylate p53. Both post-translational modifications lead to the expression of BAX, whose proapoptotic effects on the mitochondrial membrane affect the events described above. This figure was devised using BioRender icons (https://biorender.com, accessed on 26 May 2023) and the vector image bank of Servier Medical Art (https://smart.servier.com/, accessed on 26 May 2023). Servier Medical Art by Servier is licensed under a Creative Commons Attribution 3.0 Unported License.

**Figure 3 cells-12-02573-f003:**
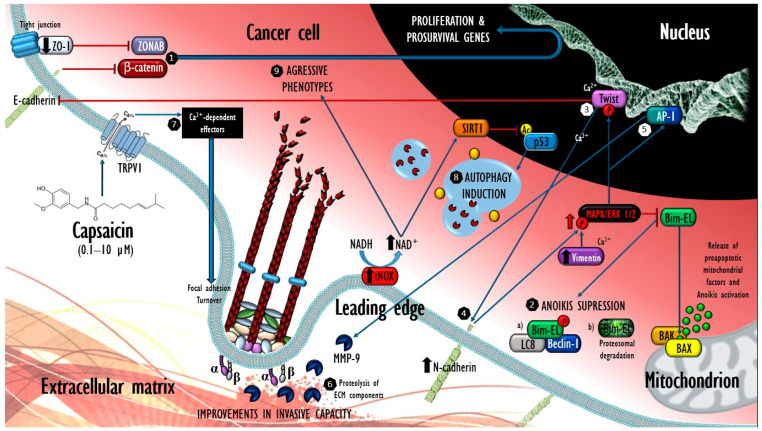
Mechanistic model of the promoter effects observed in cancer cells exposed to 0.1 to 10 µM concentrations of CAP. Improvements in proliferation and invasiveness are produced through the epithelial–mesenchymal transition. (1) After a drop in the expression of ZO-I and E-cadherin, the import of ZONAB and β-catenin to the nucleus occurs, enabling their transcriptional activity on proliferation and survival genes. The increase in MAPK/ERK 1/2 signaling phosphorylates and modulates the activity of three downstream targets, (2) Bim-EL, whose pro-anoikis activity is suppressed by (a) heterotrimer sequestration or (b) proteasomal degradation; in (3) Twist, whose transcriptional regulation would explain both E-cadherin abrogation and (4) N-cadherin overexpression, which, through its intracytoplasmic domain and in conjunction with vimentin, positively feeds back the phosphorylation of MAPK/ERK 1/2 that stimulates the expression of MMP-9 in parallel through the transcriptional activation of the AP-1 heterodimer (5), producing (6) proteolytic effects on the extracellular matrix. (7) Concomitant internalization of Ca^2+^ enabled by CAP through TRPV1 enables the cell to exhibit a rapid turnover of focal adhesions in favor of a greater invasive capacity. Finally, as a consequence of the increased expression of tNOX, the acquisition of aggressive phenotypes is stimulated (9) and, in the dependent manner of its enzymatic product (NAD^+^), SIRT1 deacetylates p53 and leads to autophagy (8), guaranteeing the availability of nutrients during the eventual circulatory dissemination of the tumor cell. Ca^2+^ dependence is noted in some of these events. This figure was devised using BioRender icons (https://biorender.com, accessed on 30 May 2023) and the vector image bank of Servier Medical Art (https://smart.servier.com/, accessed on 30 May 2023). Servier Medical Art by Servier is licensed under a Creative Commons Attribution 3.0 Unported License.

## Data Availability

Data sharing not applicable.

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
