# Peer review of "Capsaicinoids and Their Effects on Cancer: The “Double-Edged Sword” Postulate from the Molecular Scale"

_cells, 2023, doi:10.3390/cells12212573_

Round 1

Reviewer 1 Report

Comments and Suggestions for Authors

Great works have been presented in this manuscript. This review presents the origin discovery of capsaicinoids, especially capsaicin (CAP). The anti-tumor effects (selectivity, oxidative stress, and ERS) and the immunological regulation (DCs maturation, T cell activation) of CAP were collected and summarized. The authors also proposed some hypotheses potential of CAP for future clinical anti-tumor application. 

Collectively, the manuscript contains a lot of information and is eminently readable. I think the article can be published in Cells after some concerns that I list below are addressed.

There are some issues, and I have the following comments below:

1.     Line 65: The description of reference 23 was too simple, there was no obvious information on crucial results of this paper. I had to search this paper again and find indispensable results. The authors should add missing information and check this issue for full-text of the manuscript! Following the gold rule: fewer sentences, more information.

2. Figure 1: the resolution was low; there was a black bar on the right side. Please check!

3.     In the section “Molecular mechanisms behind the selective cytotoxicity of capsaicin in cancer cells.”

In general, obvious IC50 values or some specific data would be presented for discussion or results for selectivity. I did not find any description for CAP against cells (low tNOX expression).

Line 246: the concentration of CAP treatment was 100-200 μM, the concentration was too high. The time of CAP administration should be added!

4.     In the section “Immunomodulating properties of capsaicin in cancer.”

Line 965: The DAMPs-related description should be presented before the ICD-inducing related information. The tumor cells in the DAMPs process could induce the DCs maturation or T cell activation in TME. 

Comments on the Quality of English Language

Minor polishing of English language required.

Author Response

Comments and Suggestions for Authors

Great works have been presented in this manuscript. This review presents the origin discovery of capsaicinoids, especially capsaicin (CAP). The anti-tumor effects (selectivity, oxidative stress, and ERS) and the immunological regulation (DCs maturation, T cell activation) of CAP were collected and summarized. The authors also proposed some hypotheses potential of CAP for future clinical anti-tumor application. 

Collectively, the manuscript contains a lot of information and is eminently readable. I think the article can be published in Cells after some concerns that I list below are addressed.

There are some issues, and I have the following comments below:

1. Line 65: The description of reference 23 was too simple, there was no obvious information on crucial results of this paper. I had to search this paper again and find indispensable results. The authors should add missing information and check this issue for full-text of the manuscript! Following the gold rule: fewer sentences, more information.

We have incorporated appropriate descriptions of the statistical treatment of key findings in these works, as well as of the populations studied, both in reference 23, and in similar cases in references 24 and 25. Page 2, lines 62-86.

2. Figure 1: the resolution was low; there was a black bar on the right side. Please check!

Figure 1 has been replaced with a high resolution version.

3. In the section “Molecular mechanisms behind the selective cytotoxicity of capsaicin in cancer cells.”

In general, obvious IC50 values or some specific data would be presented for discussion or results for selectivity. I did not find any description for CAP against cells (low tNOX expression).

Line 246: the concentration of CAP treatment was 100-200 μM, the concentration was too high. The time of CAP administration should be added!

The possible implication that a low level of tNOX expression would cause in a scenario of low responsiveness to CAP (section 6), constitutes a hypothesis based on the mechanisms described in section 4.1, related to the preponderance of extramitochondrial oxidative stress (derived from tNOX abrogation) in CAP-responsive cells and reciprocally, how the absence of tNOX appears to explain CAP refractoriness in a variety of non-tumor cell types (section 4.1) probably due to the possibility of reestablishing cellular homeostasis through endogenous antioxidant mechanisms. Under this framework, tumor cells with low levels of tNOX expression could potentially support this latter result. As far as we have been able to verify and as you point out, there are no studies that have explored this problem and the description of the role of tNOX, in the context of responsiveness to CAP, remains in eminently dichotomous terms, so the development of studies aimed at studying this possible implication could be relevant. The exposure time from reference has been incorporated in page 6, line 268.

4.In the section “Immunomodulating properties of capsaicin in cancer.”

Line 965: The DAMPs-related description should be presented before the ICD-inducing related information. The tumor cells in the DAMPs process could induce the DCs maturation or T cell activation in TME. 

The enunciation of immunogenic cell death has been relocated after the description of DAMPs. However, the mention of externalization of CRT and HSP90 was kept at the beginning of the paragraph for reasons of sequence in the review of results. Improvements in the wording of these lines were also made. Correction was done in page 21, lines 979-987.

We greatly appreciate your observations, which have undoubtedly greatly enriched this work.

Reviewer 2 Report

Comments and Suggestions for Authors

The manuscript cells-2644163 devoted the actual field of cell and molecular biology, namely systematization of data about capsaicinoids and their effects on cancer and can be interested to the specialists working in this field. The author’s opinion is clear and based on a good literature data. I am personally impressed by the structure of the article, the systematization of scientific data and the sequence of its presentation. The paper fit the Journal scope and formal requirements. However, it needs minor revision before publication.

To improve the quality and perception of the manuscript I would suggest paying attention to following comments:

1.     It will be good to provide an expert opinion of the prospects for the further development of this scientific direction.

2.     The chemical formulas of main capsacinoids should be presented.

3.     The style of references in the Introduction section should be changed. In some cases, there are up to 10 sources after one sentence (for example, line 61 or line 85). This is unacceptable for publications in high-rated journals. Instead such references, it would be better to make a cross-reference discussion..

My decision is m
inor revision.

Comments on the Quality of English Language

Minor editing of English language required

Author Response

Comments and Suggestions for Authors

The manuscript cells-2644163 devoted the actual field of cell and molecular biology, namely systematization of data about capsaicinoids and their effects on cancer and can be interested to the specialists working in this field. The author’s opinion is clear and based on a good literature data. I am personally impressed by the structure of the article, the systematization of scientific data and the sequence of its presentation. The paper fit the Journal scope and formal requirements. However, it needs minor revision before publication.

To improve the quality and perception of the manuscript I would suggest paying attention to following comments:

1. It will be good to provide an expert opinion of the prospects for the further development of this scientific direction.

Throughout the text, the best elements we have are provided to build a well-founded perspective of the panorama of this field of study, as well as the trends for its future development. We consider that we have incorporated the work of experts in the field to define a “future trends” section.

  1. The chemical formulas of main capsacinoids should be presented.

The IUPAC nomenclature of the two main capsaicinoids have been incorporated (lines 42 and 43), as their structural formulas (Figure 1).

3. The style of references in the Introduction section should be changed. In some cases, there are up to 10sources after one sentence (for example, line 61 or line 85). This is unacceptable for publications in high-rated journals. Instead such references, it would be better to make a cross-reference discussion.

The format of the references in lines 60 and 61 and 101-106 was corrected, through a detailed differentiation of the data cited for each reference. The rest of the manuscript does not present any more references.

We greatly appreciate your observations, which have undoubtedly greatly enriched this work.